# Diet and Physical Activity in Fabry Disease: A Narrative Review

**DOI:** 10.3390/nu16071061

**Published:** 2024-04-04

**Authors:** Giovanna Muscogiuri, Oriana De Marco, Tonia Di Lorenzo, Maria Amicone, Ivana Capuano, Eleonora Riccio, Guido Iaccarino, Antonio Bianco, Teodolinda Di Risi, Antonio Pisani

**Affiliations:** 1Unità di Endocrinologia, Dipartimento di Medicina Clinica e Chirurgia, Diabetologia ed Andrologia, University of Naples Federico II, Via Sergio Pansini 5, 80131 Naples, Italy; giovanna.muscogiuri@unina.it; 2Endocrinology Unit, Centro Italiano per la Cura e il Benessere del Paziente con Obesità (C.I.B.O), Department of Clinical Medicine and Surgery, University Medical School of Naples, Via Sergio Pansini 5, 80131 Naples, Italy; toniadilorenzo55@gmail.com; 3Chair of Nephrology “Federico II”, Department of Public Health, University of Naples Federico II, Via Sergio Pansini 5, 80131 Naples, Italy; maria.amicone@unina.it (M.A.); ivanacapuano@libero.it (I.C.); lindadirisi@gmail.com (T.D.R.); or antonio.pisani13@gmail.com (A.P.); 4Institute for Biomedical Research and Innovation, National Research Council of Italy, 90146 Palermo, Italy; elyriccio@libero.it; 5Interdepartmental Research Center for Arterial Hypertension and Associated Pathologies (CIRIAPA)-Hypertension Research Center, University of Naples “Federico II”, 80131 Naples, Italy; guiaccar@unina.it (G.I.); antoniodottorbianco@gmail.com (A.B.); 6Department of Clinical Medicine and Surgery, Univeristy of Naples Federico II, 80131 Naples, Italy; 7DAI Endocrinologia, Diabetologia, Andrologia e Nutrizione Ambulatorio AFA “Attività Fisica Adattata”, University of Naples “Federico II”, 80131 Naples, Italy; 8CEINGE Biotecnologie Avanzate, Via Gaetano Salvatore 486, 80145 Naples, Italy

**Keywords:** Fabry disease, physical activity, inflammation, nutritional therapy, oxidative stress, FODMAPs

## Abstract

Fabry disease (FD) is caused by mutations in the galactosidase alpha (GLA) gene which lead to the accumulation of globotriaosylceramide (Gb-3). Enzyme replacement therapy (ERT) and oral chaperone therapy are the current pharmacological treatments for this condition. However, in the literature, there is a growing emphasis on exploring non-pharmacological therapeutic strategies to improve the quality of life of patients with FD. In particular, the nutritional approach to FD has been marginally addressed in the scientific literature, although specific dietary interventions may be useful for the management of nephropathy and gastrointestinal complications, which are often present in patients with FD. Especially in cases of confirmed diagnosis of irritable bowel syndrome (IBS), a low-FODMAP diet can represent an effective approach to improving intestinal manifestations. Furthermore, it is known that some food components, such as polyphenols, may be able to modulate some pathogenetic mechanisms underlying the disease, such as inflammation and oxidative stress. Therefore, the use of healthy dietary patterns should be encouraged in this patient group. Sports practice can be useful for patients with multi-organ involvement, particularly in cardiovascular, renal, and neurological aspects. Therefore, the aim of this review is to summarize current knowledge on the role of nutrition and physical activity in FD patients.

## 1. Introduction

Fabry disease (FD) is a rare genetic disorder characterized by a deficiency of the enzyme α-galactosidase A (α-GAL A), causing the accumulation of glycosphingolipids in various tissues and organs [1]. The prevalence of FD in Europe varies from 1 in 3100 to 1 in 117,000 [2], and it has been documented in diverse ethnic groups without discernible predilection to date [3]. The precise prevalence of FD remains a subject of ongoing debate. Screening studies, including both neonatal and dialysis assessments, have revealed a higher prevalence than previously estimated. In particular, the identification of late-onset phenotypes has contributed to a delay in diagnosis [4,5].

The implementation of multiple strategies aimed at improving symptoms and managing complications has led to an increased life expectancy for patients with FD in recent years [6]. This, in turn, emphasizes the growing importance of addressing patients’ comorbidities and lifestyle in their overall care. There are currently two effective treatments for FD: enzyme replacement therapy (ERT) and chaperone therapy [6].

In addition to pharmaceutical treatment and systemic diseases, physical activity (PA) and nutrition have been demonstrated to play a significant role in determining the pathological outcome [7,8,9,10].

Some recent studies have explored the connection between nutrition and illness, proposing the use of dietary approaches for effective disease management [11,12].

This review aims to offer a comprehensive description of diet and PA guidelines tailored specifically for patients with FD and its associated complications. The intention is to establish a solid foundation for a thorough exploration of these aspects, recognizing their profound influences on patients’ lives and emphasizing their critical role in the overall multicentric care of individuals with FD.

## 2. The Genetics of Fabry Disease

FD is an X-linked recessive hereditary disorder resulting from mutations in the galactosidase alpha (GLA) gene [6,13]. A single mutation in this gene in FD can result in a variety of clinical presentations, adding complexity to the complex link between genotype and phenotype [14].

There has been a proposal to classify FD into two main phenotypes: “classic” and “late onset” subtypes. Individuals exhibiting the classic FD phenotype typically display minimal or no residual enzyme activity, experiencing the onset of acroparesthesia, hypohidrosis, angiokeratomas, and/or a distinctive corneal dystrophy during childhood or adolescence [14]. Clinical manifestations in heterozygous females range from asymptomatic during a normal life to severity comparable to that of many affected males [15]. The diversity in the clinical presentations of women with FD has been attributed to random X chromosome inactivation [16,17]. The α-GAL enzyme, comprising approximately 429 amino acids, plays a crucial role in cleaving glycosphingolipid globotriaosylceramide (Gb3) into galactose and lactosylceramide within lysosomes [18]. Consequently, in individuals with FD, there is an accumulation of Gb3 in various tissues. This accumulation tends to favor the vascular endothelium and smooth muscle cells of the cardiovascular system, as well as renal podocytes, providing an explanation for the predominant clinical manifestations observed in these organs [18].

## 3. Organ Involvement in Fabry Disease

The intracellular accumulation of globotriaosylceramide in FD induces organ damage, contributing to a systemic pathological condition.

1. Skin: Angiokeratomas, which are tiny, dark-red lesions on the skin, are a common sign of skin involvement in FD. Diffuse angiokeratomas should alert the physician to a possible diagnosis of FD [19,20].

2. Kidneys: The kidney is one of the primary target organs. The classic phenotype male exhibits hyperfiltration and microalbuminuria as early indicators of renal involvement, which progresses to chronic kidney disease (CKD) [21,22,23,24].

3. Heart: The cardiovascular aspect of FD typically presents with left ventricular hypertrophy, myocardial fibrosis, heart failure, hypertension, and arrhythmias.

These conditions not only restrict the quality of life but are also the most prevalent causes of mortality [25,26].

4. Nervous system: FD can impact the nervous system, causing neurological symptoms such as neuropathic pain, stroke, and increased susceptibility to lesions of the cerebral white matter [27,28].

5. Gastrointestinal system: the gastrointestinal tract may be affected, causing problems such as abdominal pain, nausea, and diarrhea [29].

Treatment for FD includes ERT, utilizing agalsidase alfa and beta, which are distinct medications composed of recombined proteins employed in ERT, a therapeutic approach aiming to introduce the deficient enzyme into the patient’s system [1,30,31].

An additional therapeutic option for FD involves oral chaperone therapy, specifically recommended for patients with “susceptible” mutated forms of α-galactosidase A (GLa variants) [32,33].

This therapy has the benefit of being administered non-invasively, but it also has the drawback of being an oral therapy that depends on patient compliance [34], as seen in other nephropathic patients [35]. The introduction of ERT has transformed the management of FD, providing significant benefits to patients in multiple aspects, such as reduction of disease progression, improvement of cardiac function, renal protection, and reduction of pain and other symptoms [36,37,38,39].

## 4. Nutritional Aspects of Management of CKD in Fabry Patients

FD affects the kidney in both male and female patients, leading to end-stage renal disease (ESRD) and premature death [40]. Along with ERT, several strategies are needed to prevent and treat FD nephropathy and its complications, such as cardiovascular events, hypertension, and bone disorders [40]. In terms of dietary therapy, patients with FD and renal complications should follow similar recommendations to those for patients with renal dysfunction as glomerular hyperfiltration, increased proteinuria, and accumulation of protein waste products occur in both subjects [41]. Low-protein diets have been a therapeutic intervention for years to slow the progression of CKD towards kidney failure as they reduce glomerular hyperfiltration, proteinuria, and the accumulation of protein waste products [42].

Indeed, in humans with pre-existing kidney failure, high-protein diets have been shown to accelerate the decline of kidney function [43], as high dietary protein can cause intraglomerular hypertension with consequent renal hyperfiltration, glomerular damage, and proteinuria [44]. On the contrary, a low-protein diet reduces nitrogen waste products and the renal workload, and this leads to a lowering of intra-glomerular pressure and an improvement in outcomes related to total renal failure [43,44]. Therefore, there is general agreement on the effect of dietary protein restriction on reducing the rate of glomerular filtration rate (GFR) loss in most forms of renal failure [45]. Furthermore, low-protein diets can attenuate insulin resistance and oxidative stress, the latter representing an important trigger in the development of organ damage in FD patients [46]. Both insulin resistance and oxidative stress may play a role in accelerating atherosclerosis among CKD patients with or without Fabry disease [46].

However, protein restriction is not without complications, as there is a risk of negative nitrogen balance, weight loss, and malnutrition [40]. For this reason, it has been suggested that the amount of protein intake should be based on nutritional requirements for age and sex. To avoid protein-energy waste (PEW), an adequate quality of protein and energy intake must therefore be guaranteed [40].

Regarding protein intake, the Kidney Disease Outcomes Quality Initiative (KDOQI) guidelines recommend a low-protein diet providing 0.55–0.6 g/kg/day for patients with stage 3–5 CKD not on dialysis who are metabolically impaired but stable, or a very low-protein diet providing 0.28–0.43 g/kg/day plus the use of keto analogues [47]. For the early stages (1–3) of the disease, there is no clear recommendation, although some guidelines suggest the use of moderately low-protein diets equal to 0.8–1 g protein/kg body weight/day [47]. In order to maintain an adequate nutritional status, considering age, sex, PA level, body composition, weight goals, stage of CKD, and presence of concomitant disease or inflammation, an energy intake of 25–35 kcal/kg body weight per day should be achieved [47]. Furthermore, there appears to be a different impact on CKD as determined by the dietary protein source (animal or plant) [48]. Indeed, animal proteins are strongly associated with clinical features of diabetic kidney disease (DKD), such as glomerular hyperfiltration, albuminuria, and a decline in renal function. In contrast, plant-based proteins have a strong beneficial effect on both DKD and cardiovascular disease (CVD) [48]. Specifically, potential mediators of animal protein-induced kidney damage include dietary acid load, phosphate content, gut microbiome dysbiosis, and resulting inflammation [44].

Considering this, much recent scientific evidence seems to recommend plant-based diets for both primary and secondary prevention of CKD [49,50,51,52]. Regarding salt intake, extensive research has shown that salt restriction has great potential to slow the progression of kidney disease and mitigate serious complications of CKD, such as high blood pressure [40]. For this reason, the 2021 Kidney Disease Outcomes Quality Initiative (KDIGO) Clinical Practice Guidelines for the Management of Blood Pressure in CKD suggest aiming for a sodium intake of <2 g sodium per day (or <90 mmol sodium per day, or <5 g sodium chloride per day) in patients with high blood pressure and chronic renal failure [53].

It is also noteworthy that the beneficial effects of salt restriction may go far beyond reducing blood pressure [54]. A low-sodium diet, in fact, can itself reduce proteinuria and albuminuria, as well as oxidative stress, inflammation, and endothelial cell dysfunction [54]. Therefore, in order to achieve a restriction of salt in the diet, it might be helpful to suggest that the patient limit the salt added to foods, preferring the use of spices to enhance the flavor of foods, and, finally, avoid ultra-processed salty products such as canned foods, potato chips, cured meats, and aged cheeses.

Regarding the intake of phosphorus and potassium, it is important to remember that the KDOQI guidelines do not recommend restricting the intake of phosphorus and potassium unless the biochemical values of the blood are altered [47]. Thus, in the case of CKD, the goals of nutritional management are to provide energy and micronutrient requirements, preserve renal function, normalize body fluid composition, and minimize hyperphosphatemia and renal osteodystrophy [40]. Given the influence of renal complications on FD, these dietary goals should also be pursued and implemented in nephropathic patients with FD.

## 5. Nutritional Approaches for the Management of Irritable Bowel Syndrome in Patients with Fabry Disease 

Many patients with FD experience intense gastrointestinal symptoms, which appear to have a negative impact on quality of life [41]. In fact, a comparative study conducted on 342 patients with FD verified that adult patients with gastrointestinal symptoms had a significantly lower quality of life score, as assessed by the EQ-5D questionnaire, compared to patients without gastrointestinal symptoms [36]. Among the most common gastrointestinal symptoms found in patients suffering from FD is abdominal pain [40]. The latter affects up to a third of patients, worsens after meals and in the presence of a stress condition, and usually appears in the early stages of the disease. Patients in these cases report swelling, cramps, burning pain, and abdominal discomfort [40]. The second most common gastrointestinal symptom is diarrhea, which occurs in 20% of patients [55]. Although diarrhea is a common complaint, there is a subgroup of patients, mainly women, who suffer from constipation, which is often debilitating. Slightly less common, but still significant, are reports of symptoms such as vomiting, nausea, and early satiety. These sensations can lead the patient to serious and dangerous dietary restrictions due to fear of ingesting food, resulting in a loss of body weight [55]. The mechanisms that cause the gastrointestinal symptoms of FD are complex, multifactorial and not yet fully elucidated [32]. In general, it is believed that the accumulation of Gb3 in enteric neurons and vessels and the resulting inflammation can cause dysregulation of the autonomic nervous system, with impairment of enteric neurons responsible for the control of intestinal motility [32,40]. At the same time, the slowing of intestinal transit leads to an excessive growth of intestinal bacteria, which amplifies typical gastrointestinal symptoms. In addition to poor intestinal motility, recent studies suggest that globotriaosylsphingosine (lyso-Gb3) directly affects microbial growth, which could lead to dysbiosis, an imbalance of intestinal flora [32].

Furthermore, it must be considered that gastrointestinal symptoms in patients with FD could be due to the lack of digestion of oligosaccharides within the intestine, given that the presence of the enzyme α-galactosidase A (α-GAL A) is necessary for such digestion [32].

Regarding current therapies, ERT appears to improve gastrointestinal symptoms associated with FD [40]. However, this therapy is not always effective; in fact, more than half of patients treated with ERT continue to have gastrointestinal symptoms, and some patients even develop new gastrointestinal symptoms during treatment with ERT [40]. Therefore, in order to improve the quality of life and reduce the morbidity rate of patients suffering from FD and gastrointestinal complications, it is necessary to institute a specific intervention in which pharmacological therapy is also accompanied by dietary therapy. Diarrhea and abdominal pain seen in patients with FD are known to mimic irritable bowel syndrome (IBS), with diarrhea predominating [37]. However, the pathophysiological similarities between the two pathologies need to be clarified [41].

Because symptoms are often similar to those seen in IBS, it may be helpful for the Fabry disease patient suffering from these gastrointestinal complications to follow the dietary recommendations provided for IBS [55].

The latter is defined as a chronic and debilitating gastrointestinal disease characterized by altered bowel habits associated with abdominal discomfort or pain in the absence of detectable structural and biochemical abnormalities [56]. In this condition, some foods, such as spicy foods, caffeine, alcohol, insoluble fiber, fats, and spices are not recommended and therefore should also be avoided in patients suffering from FD who simultaneously present gastrointestinal complications that overlap with those of IBS [29]. Some mechanisms could explain why these food components amplify IBS symptoms [57].

For example, alcohol is known to affect gastrointestinal motility, absorption, and intestinal permeability, while caffeine increases gastric acid secretion and colonic motor activity as well as rectosigmoid motor activity. As regards spicier foods, it should be considered that capsaicin, an active component of chili pepper, is responsible for accelerating gastrointestinal transit and could cause abdominal pain and a burning sensation in healthy subjects and those with IBS. Finally, fat stimulates the gastrocolic reflex, and if administered directly into the duodenum, the response is prolonged and exaggerated in subjects suffering from IBS [57]. However, the literature recommends carefully evaluating the intake of these food components in patients with IBS and limiting their intake only when this intake is related to IBS symptoms [58]. The role of individual FODMAPs in Fabry disease has not been investigated, except in cases of co-presence with IBS, but it is known that these FODMAPs require the enzyme α-GAL A, which is deficient in such patients. Therefore, it is possible that many patients with Fabry disease have difficulty digesting these nutritional compounds, resulting in gastrointestinal symptoms.

Currently, the British Dietetic Association (BDA) guidelines for the dietary management of IBS in adults recommend a specific dietary intervention, such as a diet low in fermentable oligosaccharides, disaccharides, monosaccharides, and polyols (FODMAP), in cases where IBS symptoms persist, despite implementation of diet and lifestyle recommendations [58].

In general, FODMAPs are a broad class of small carbohydrates containing only 1–10 sugars, generally found in a range of very common and diverse foods (Table 1) [59].

They are also classified into four main groups depending on the length of the carbohydrate chain and intestinal fermentation processes [60,61]. Among the oligosaccharides, we find fructo-oligosaccharides (FOS) and galacto-oligosaccharides (GOS) present in wheat and rye-based products, legumes, nuts, artichokes, onions, and garlic. As a disaccharide considered FODMAP, there is lactose, the sugar present in dairy products that requires the lactase enzyme for its absorption [60].

Another carbohydrate considered poorly absorbable is fructose, the smallest FODMAP carbohydrate, present especially in some fruits, including apples, pears, watermelon, and mango, as well as in honey and some vegetables. Finally, there is the family of polyols, widely used as artificial sweeteners, for example, in chewing gum and sugar-free mints. Among these, we remember those most commonly present in foods, namely mannitol and sorbitol [60]. Being poorly absorbed in the small intestine, FODMAPs can pass without being absorbed into the colon, where they increase water in the lumen through osmotic activity and induce the production of gas due to fermentation by colon bacteria [58]. This fermentation may cause luminal distention and gastrointestinal symptoms in IBS patients.

Based on the role of FODMAPs in the pathogenesis of IBS, the low-FODMAP diet limits dietary intake of these carbohydrates with the overall goal of improving IBS symptoms [58].

The dietary protocol in question consists of two different phases: the first generally lasts 4–6 weeks and involves the total elimination of foods containing FODMAPs, while the second consists of the progressive consumption of small quantities of foods containing FODMAPs [61].

This last phase lasts between 8 and 10 weeks [61]. Finally, a personalization of the diet will be implemented for longer-term use. Observational studies and randomized trials have indicated that the low-FODMAP dietary approach leads to symptom improvement in up to two-thirds of IBS patients [62,63,64,65]. Therefore, it has been hypothesized that a low-FODMAP diet may benefit patients with FD and gastrointestinal complications associated with IBS [41] (Figure 1).

Therefore, in cases where Fabry disease and IBS coexist, the use of a dietary protocol with low-FODMAP foods could be considered. Such a protocol, if it is tailored to the patient and applied properly, could be helpful in reducing gastrointestinal symptoms such as diarrhea and abdominal bloating.

It is conceivable that FD patients in particular may benefit from this dietary approach since many FODMAPs require the enzyme α-GAL A for correct digestion [66]. Currently, in the literature, there is only one study that evaluates the effectiveness of a low-FODMAP protocol in patients with FD and gastrointestinal symptoms [61]. This retrospective study demonstrated that the low-FODMAP diet was effective in reducing gastrointestinal symptoms such as indigestion, diarrhea, and constipation in seven adults with a mean age of 47 years and with FD with severe gastrointestinal manifestations [61]. Furthermore, the foods that patients indicated as being most involved in gastrointestinal symptoms were those rich in gluten and lactose and some vegetables, mainly from the Brassicacae and Lyliaceae species. However, it must be considered that this result was obtained in the context of a very small number of subjects. Furthermore, it should not be forgotten that a low-FODMAP diet is not easy to follow, and some patients may not adhere to this treatment or abandon it along the way [61]. Moreover, limiting the dietary intake of prebiotic fructans and galacto-oligosaccharides can alter the luminal microbiota [58]. Therefore, to obtain positive results from the application of this protocol, it is advisable to seek dietary advice from a specialized dietician [58]. The latter will be able to provide a detailed dietary plan so that the low-FODMAP diet is nutritionally adequate as well as personalized to the patient’s needs. In any case, it must be considered that not all hospital nephrology departments have a renal dietitian available. In fact, recent surveys highlight a critical workforce shortage of dieticians in the global kidney health sector [67]. Considering the relevant presence of intestinal symptoms in subjects with FD and the potential role of dietary interventions, further clinical studies should be encouraged regarding the impact of the low-FODMAP diet on gastrointestinal manifestations in larger patient cohorts with FD, as well as the development of further useful strategies to increase patient compliance with this protocol [61]. Therefore, given the complexity of FD, we believe it is appropriate that in the team that deals with the multidisciplinary management of this pathology, in addition to specialist doctors such as cardiologists, psychologists, nephrologists, geneticists, and gastroenterologists, there are also specialized dieticians capable of supporting the patient from a strictly nutritional point of view.

Furthermore, in FD, among the various gastrointestinal manifestations that may occur, there are also a sense of postprandial fullness and early satiety [40]. These manifestations are part of a set of symptoms that lead to functional dyspepsia. In addition to drug therapy with prokinetic agents, approaches to dyspepsia include dietary advice such as ingestion of small, regular, low-fat meals [40]. Such advice should also be extended to patients with FD in the presence of an established diagnosis of functional dyspepsia.

## 6. Nutritional Components and Dietary Models Useful in Modulating Oxidative Stress and Inflammation in Patients with Fabry Disease

Oxidative stress (OS) is widely considered to be involved in the cardiovascular and renal complications of FD [68].

Although no specific studies on FD are available, it is known that diet is one of the factors that regulates the cellular response and genes involved in the development of oxidative stress [40,69].

The Western diet, now widespread in the general population, involves a marked and high consumption of ultra-processed foods, simple carbohydrates, and saturated fatty acids (SFA), and a lack of natural foods rich in antioxidants [69]. In the long run, these unhealthy habits can lead to an excessive production of pro-inflammatory and pro-oxidant agents and a decrease in the molecules that neutralize these agents, i.e., antioxidants [69]. In this regard, specific dietary interventions and bioactive molecules present in foods can interfere with some pathways, leading to oxidative stress [68]. Currently, an antioxidant treatment for FD has been marginally studied [68]. Only one study in the literature has evaluated the role of the metabolism of an antioxidant, namely glutathione (GSH), in FD. In fact, Kim et al., using GLA-mutant renal organoids, an in vitro disease model of renal FD, demonstrated that in the latter there is a decrease in the metabolism of the antioxidant GSH [70]. Furthermore, it was seen that treatment with GSH reduced OS and attenuated structural alterations in this model [70]. This result is interesting given that a reduction in antioxidant defenses in FD is also documented by the reduced activity of GSH and GSH peroxidase and by the increase in superoxide dismutase–catalase ratios in the erythrocytes of FD patients subjected to ERT [40]. Another antioxidant molecule that appears to have implications for the pathophysiology and pathogenesis of FD is vitamin C. Ifechukwude Ebenuwa et al. studied the prevalence, clinical characteristics, and genomic associations of renal vitamin C loss in FD and found that, compared to controls (34 subjects), the Fabry cohort (33 subjects) was 16 times more likely to suffer renal loss of vitamin C [71]. However, prospective longitudinal studies are needed to investigate the long-term clinical outcomes of renal vitamin C loss in FD and the efficacy of early vitamin C supplementation [71].

Polyphenols also contribute to counteracting oxidative stress, and are contained in fruit, vegetables, chocolate, and tea [72].

In particular, green tea contains high quantities of flavonoids, such as catechins. Among these, the most abundant is epigallocatechin-3-gallate (EGCG), which has been shown to exert an anti-inflammatory function in several animal models of acute kidney injury, glomerulonephritis, lupus nephritis, diabetic nephropathy, and renal fibrosis [68]. Furthermore, Giovanni Bertoldi et al. hypothesized in their study a possible role for the administration of green tea as an adjuvant therapy to ERT in patients with FD.

From the results of the study conducted on ten patients with FD, it was demonstrated that the antioxidant effect exerted by ERT itself was further amplified by the adjuvant treatment with green tea [68]. This highlights the fundamental importance of early treatment for FD while also underscoring the likely positive effect of an adjuvant antioxidant treatment toward the reduction of OS [68].

In FD, together with oxidative stress, chronic inflammation can also lead to irreversible tissue damage with target organ fibrosis [40]. Gb3 and lyso-Gb3 accumulation and their identification as a danger signal are the main causes of inflammation and the innate immune system’s simultaneous activation that accompany FD [23]. Chronic inflammation in FD results from ongoing exposure to glycolipids, and it is possible that tissue damage advances on its own after the initial inflammatory reaction to Gb3 deposition [23].

It is unclear exactly which cellular and molecular mechanisms connect organ pathology and inflammatory processes to the intracellular build-up of substrates such as Gb3 [23].

Among the dietary models useful for counteracting the oxidative and inflammatory processes of cells and avoiding DNA damage, there is certainly the Mediterranean diet [73]. Paying attention to the relationship between specific foods and inflammation, the protective effects of the Mediterranean diet can be attributed to the high concentration of polyphenols contained in olive oil, wine, and vegetables, all foods known for their antioxidant and anti-inflammatory capacity [73].

Among the polyphenols, we remember those present in olive oil, such as oleuropein, hydroxytyrosol, and tyrosol, which are able to exert antioxidant and anti-inflammatory activity, with consequent improvement of oxidative stress [74]. In conclusion, it should be considered, however, that it is not known with certainty which foods or nutrients in the Mediterranean diet are responsible for the anti-inflammatory effect, but accumulated data suggest that the synergy between nutrients coming from a range of different foods may play a role in reducing inflammation [75].

Indeed, the effect of individual dietary components may be too small to detect, but their additive impact may be large enough to detect.

In support of this hypothesis, data from numerous epidemiological studies suggest that people who consume higher-quality diets have lower inflammation, regardless of classic cardiometabolic risk factors [75,76,77,78]. Therefore, it could be plausible to use antioxidant and anti-inflammatory dietary patterns as a useful complement to classical FD therapy in order to slow down the pathogenetic mechanisms that underlie the progression of FD [40].

## 7. Physical Activity in Fabry Disease

Few published studies have been done on exercise in FD patients. In general terms, it is well recognized how essential PA is for preserving and enhancing muscular fitness and the cardiovascular system, as well as how crucial it is to manage these variables in systemic diseases [79].

Given the importance of cardiac and renal involvement in FD patients, it is reasonable to assume that these advantages would also extend to them (Figure 2).

In patients with Fabry disease with renal complications, monitoring of nutritional status and control of the intake of certain nutrients, such as protein, salt, phosphorus, and potassium, should be carried out. In addition, the use of vegetable protein at the expense of animal protein should be encouraged. On the other hand, the Fabry patient with a diagnosis of IBS might benefit from the use of a low-FODMAP diet. Physical activity, performed under medical supervision, is equally important in the management of a patient with Fabry disease and can be beneficial for cardiovascular, renal, and neurological aspects.

The advantages of even brief daily PA sessions are emphasized in the eleventh edition of the American College of Sports Medicine’s (ACSM) Guidelines for Exercise Testing and Prescription [80]. Health problems resulting from insufficient PA are similar to those caused by obesity and smoking. Frequent movement not only enhances mood and sleep but also reduces the risk of many diseases, delays the aging process, and strengthens the immune system. Furthermore, maintaining an active lifestyle lowers the risk of dementia and Alzheimer’s disease and enhances mental acuity as one ages [80]. Interestingly, the ACSM guidelines require that patients with a known diagnosis of a chronic condition, as well as those suspected to be diagnosed with a chronic condition, must be evaluated before the prescription of a physical exercise program. Indeed, the cardiovascular risk of a patient should be assessed in order to avoid the prescription of physical exercise causing harm to the patient [80]. The prescription of PA or exercise to an FD patient therefore should consider this piece of advice closely. Indeed, patients with FD present advanced organ damage such as left ventricle hypertrophy as well as reduced kidney function, which, per se, are two determinants of increased cardiovascular risk. Furthermore, neuropathic damage, as well as reduced sweating and impaired muscular metabolism, might impact the tolerance to stress of FD patients. The Guidelines for the Prevention, Detection, Evaluation, and Management of High Blood Pressure in Adults establish that adults with elevated blood pressure or hypertension should engage in a structured exercise program along with increased PA [81]. According to the KDIGO guidelines for 2021, every patient with kidney disease should participate in at least 150 min of PA each week.

Individualized attention should be given to the type and intensity of exercise, as even at lower than recommended levels of exercise, there may be health benefits [82]. Individuals with CKD often experience heightened cardiovascular morbidity and mortality, reduced health-related quality of life, and challenges in performing physical tasks.

Moderate-intensity exercise has proven effective in enhancing metabolism and cardiovascular health for both healthy individuals and CKD patients [83]. Long-term exercise training is recommended as a crucial component in managing various chronic conditions.

A randomized controlled pilot study conducted on pre-dialysis CKD patients suggests that such exercise regimens may lead to improvements in arterial stiffness, physical impairment, and overall health-related quality of life in this patient group [83]. Furthermore, according to the European Society of Cardiology (ESC) 2020 guidelines, the presence of a cardiomyopathy must be considered when prescribing physical exercise.

In fact, although there is limited evidence regarding the risks to subjects who undertake this activity, a careful cardiological evaluation before and during the follow-up of PA is essential [84].

Considering the cardiovascular profile, it is advisable for PA to be regulated by a prescription before starting sports and exercises.

In conclusion, patients with FD, like other nephropathic and cardiological patients, should carry out regular PA according to the indications of KDIGO 2021 and ESC 2020 but with precautions and under prescription.

## 8. Discussion

To date, due to the numerous advantages that PA has for both mental and physical health, it is evident that individuals with FD should lead active lives, but they should exercise with caution. A dietary approach in the treatment of Fabry disease (FD) has been studied to a limited extent, although it may be a useful adjunct to FD therapy [40]. It is known that patients suffering from Fabry disease complicated by chronic kidney disease (CKD) should pursue the same nutritional objectives recommended for patients with CKD, namely: providing the right energy and micronutrient needs, normalizing the composition of body fluids, minimizing hyperphosphatemia and hyperkalemia, and prefering vegetable protein sources to animal ones. Currently, the low compliance of CKD patients with low-protein diets represents a major obstacle to the optimal success of this nutritional therapy; therefore, an objective for future research could concern the identification of strategies capable of increasing compliance. In the case of patients with Fabry disease and gastrointestinal symptoms associated with irritable bowel syndrome (IBS), such as diarrhea, cramps, bloating, and abdominal pain, a protocol low in fermentable oligosaccharides, disaccharides, monosaccharides, and polyols may be useful (FODMAP). This protocol has proven useful in calming gastrointestinal symptoms, which are often disabling and force the patient to miss work or give up instances of socializing. Furthermore, this protocol could reduce intestinal dysbiosis, i.e., a condition of imbalance in the composition and functionality of the intestinal bacterial flora, which can often be present in patients suffering from Fabry disease. However, it should be considered that the long-term results of the low-FODMAP diet have not been evaluated in any study, and this represents a serious gap in scientific research. Therefore, a future goal will be to require longer follow-up for large cohorts of patients with Fabry disease. We expect that the search for long-term results of this protocol will significantly improve gastrointestinal symptoms in patients with Fabry disease who present these complications, leading to a reduction in healthcare costs and an improvement in quality of life, thus lightening the burden of the disease. Furthermore, we believe that a possible future direction of research is to identify the specific dietary habits of patients with Fabry disease through the use of widely used methods such as 3- or 7-day dietary records or 24-h dietary recalls. This information could be useful to formulate more precise and personalized dietary advice for the patient and to intercept any nutritional deficiencies, filling them quickly. We also believe it is necessary to extend nutritional evaluation to all patients affected by Fabry disease, which includes periodic measurement of the patients’ height and weight, body mass index, evaluation of body composition, and laboratory tests useful for evaluating nutritional status. This information provides the doctor with a more complete framework for the clinical–nutritional evaluation of the patient, allowing them to delay or prevent future complications. Furthermore, systematically evaluating the patient’s nutritional status and ensuring a better one will allow the ongoing pharmacological therapy to be enhanced, making it more effective and functional for the patient. Moreover, the role that diet plays in the remodulation of oxidative and inflammatory stress, which is very present in Fabry disease and responsible for the renal and cardiovascular complications of the disease, should not be forgotten. Therefore, a further area of future research could concern the impact of specific nutrients capable of counteracting both inflammation and oxidative stress in Fabry disease, as well as the use of possible dietary models capable of modulating these phenomena and positively influencing the intestinal microbiota of patients suffering from Fabry disease. Future scientific research on these topics could lead to the identification and implementation of useful tools in daily clinical practice. Current scientific research on Fabry disease and non-pharmacological treatment options such as diet and physical activity has some biases. The main ones concern the low number of studies conducted, the lack of long-term effects assessed, and the small number of subjects included in the studies cited. Therefore, to strengthen the body of knowledge regarding the impact of nutrition and exercise on individuals with FD, further research that also takes these biases into consideration is needed.

## 9. Conclusions

The results of this research highlight the value of a multidisciplinary approach for the treatment of patients with FD in offering patients with systemic diseases the best cost-effective follow-up. Additionally, it is important to determine which dietary traits and behavioral approaches work best in combination with current pharmaceutical therapies to slow the progression of FD. Therefore, current public health challenges will require the formulation of new guidelines for the nutritional management of FD.

## Figures and Tables

**Figure 1 nutrients-16-01061-f001:**
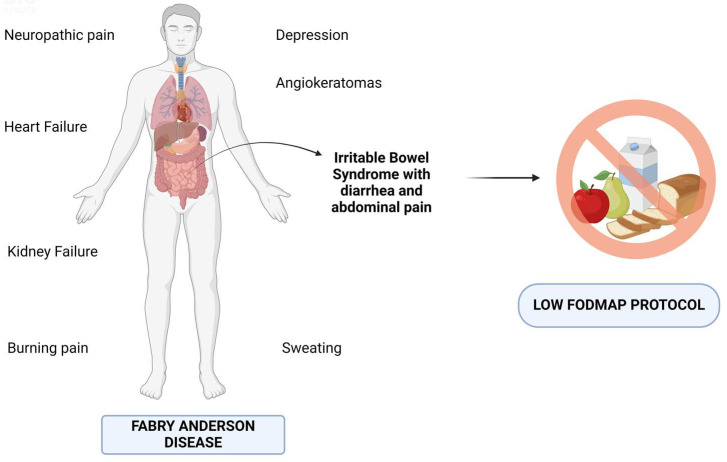
Dietary approach to implement for the management of irritable bowel syndrome in patients with Fabry disease. Fermentable Oligosaccharides, Disaccharides, Monosaccharides, and Polyols, FODMAP.

**Figure 2 nutrients-16-01061-f002:**
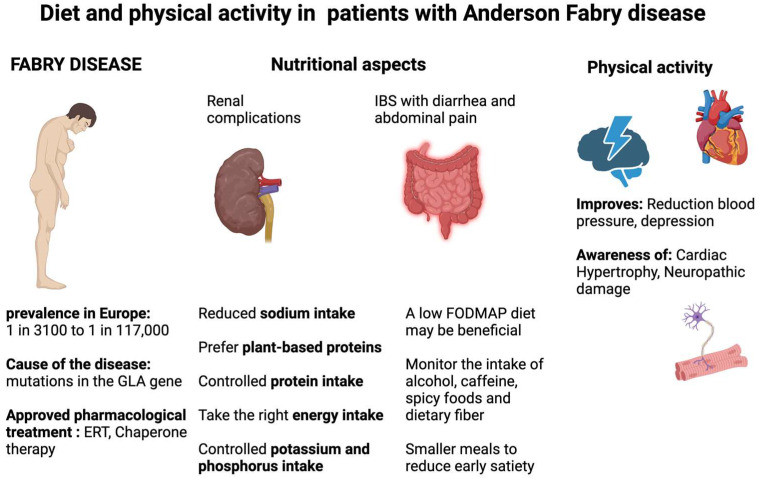
Dietary recommendations and effects of physical activity in patients with Fabry disease. Enzyme replacement therapy, ERT; galactosidase alpha, GLA.

**Table 1 nutrients-16-01061-t001:** The main food sources of FODMAPs. FODMAP: fermentable oligosaccharides, disaccharides, monosaccharides, and polyols.

Categories of FODMAPs	Main Food Sources
*Oligosaccharides:* Fructo-oligosaccharides (FOS) Galacto-oligosaccharides (GOS)	wheat products (bread/pasta/breakfast cereals), some vegetables (e.g., onion, garlic, artichoke, beetroot) Legumes: chickpeas, soybeans, lentils, red kidney beans
*Disaccharides:* Lactose *Monosaccharides:* Fructose *Polyols:* Sorbitol Mannitol Lactitol, xylitol, erythritol, maltitol	dairy products such as milk, soft cheeses, yogurt, ice-cream, sweetened condensed milk apples, pears, watermelon, mango, honey, corn syrup apples, pears, blackberries, sugar-free gum mushroom, cauliflower, sugar-free gums sugar-free gum

## Data Availability

Not applicable.

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
