# Peer review of "Diet and Physical Activity in Fabry Disease: A Narrative Review"

_nutrients, 2024, doi:10.3390/nu16071061_

Round 1

Reviewer 1 Report

Comments and Suggestions for Authors

1- Simplifying complex genetic discussions could make the article more accessible.

2- Adding patient case studies might also help illustrate practical application.

3- Highlighting specific areas for further research, particularly on long-term outcomes of the suggested interventions, would be beneficial.

Author Response

Dear Reviewer,

Reviewer 2 Report

Comments and Suggestions for Authors

The authors have done a good job explaining a relatively rare disease such as Fabry's disease and available modalities for managing the condition. 

The global prevalence of Fabry's disease should also be included in the introduction section.

The rationale for low protein diets in Fabry's disease needs more elaboration!

The role of FODMAP in Fabry's disease needs to be discussed first before discussing its role in managing the complications of Fabry's disease such as IBS!

The discussion section rather appears like the conclusion section and needs a heading change!

Other than these minor recommendations, in general, this narrative review is very nicely written and well-organized!

Author Response

Dear Reviewer,

please see the attachmenrt

Reviewer 3 Report

Comments and Suggestions for Authors

I think this is a great review.

However, I think that the discussión should be expanded, showing bias and future directions.

In addition, I would ocmplete figure 2 including pharmacological approach for the Fabry disease.

Author Response

Dear Reviewer,
